# Empathy and its associations with age and sociodemographic characteristics in a large UK population sample

**Andrew Sommerlad** [1,2]*, **Jonathan Huntley**[1,2], **Gill Livingston**[1,2], **Katherine P. Rankin**[3], **Daisy Fancourt**[4]

**1** Division of Psychiatry, University College London, London, United Kingdom, **2** Camden and Islington NHS Foundation Trust, London, United Kingdom, **3** Department of Neurology, Memory and Aging Center, University of California, San Francisco, California, United States of America, **4** Department of Behavioural Science and Health, University College London, London, United Kingdom

* a.sommerlad@ucl.ac.uk

**Data Availability Statement:** The data files supporting the findings of this study are available on request from the study data access committee

## Abstract

### Objectives

Empathy is fundamental to social cognition, driving prosocial behaviour and mental health but associations with aging and other socio-demographic characteristics are unclear. We therefore aimed to characterise associations of these characteristics with two main self-reported components of empathy, namely empathic-concern (feeling compassion) and perspective-taking (understanding others' perspective).

### Methods

We asked participants in an internet-based survey of UK-dwelling adults aged ≥18 years to complete the Interpersonal Reactivity Index subscales measuring empathic concern and perspective taking, and sociodemographic and personality questionnaires. We weighted the sample to be UK population representative and employed multivariable weighted linear regression models.

### Results

In 30,033 respondents, mean empathic concern score was 3.86 (95% confidence interval 3.85, 3.88) and perspective taking was 3.57 (3.56. 3.59); the correlation between these sub-scores was 0.45 (p < 0.001). Empathic concern and perspective taking followed an inverse-u shape trajectory in women with peak between 40 and 50 years whereas in men, perspective taking declines with age but empathic concern increases.

In fully adjusted models, greater empathic concern was associated with female gender, non-white ethnicity, having more education, working in health, social-care, or childcare professions, and having higher neuroticism, extroversion, openness to experience and agreeableness traits. Perspective taking was associated with younger age, female gender, more education, employment in health or social-care, neuroticism, openness, and agreeableness.

through data sharing agreement via f.bu@ucl.ac.
uk. The full data is not currently publicly available
due to funding arrangements but will be made
publicly available following completion of the
COVID-19 Social Study at the start of 2022.

**Funding:** The Covid-19 Social Study was funded by
the Nuffield Foundation [WEL/FR-000022583], but
the views expressed are those of the authors and
not necessarily the Foundation. The study was also
supported by the MARCH Mental Health Network
funded by the Cross-Disciplinary Mental Health
Network Plus initiative supported by UK Research
and Innovation [ES/S002588/1], and by the
Wellcome Trust [221400/Z/20/Z]. AS is funded by
the UCL / Wellcome Trust Institutional Strategic
Support Fund [204841/Z/16/Z] and by the
University College London Hospitals' (UCLH)
National Institute for Health Research (NIHR)
Biomedical Research Centre (BRC). JH is funded
by a Wellcome Trust Clinical Research Career
Development Fellowship [214547/Z/18/Z] and
supported by UCLH NIHR BRC. GL is supported by
UCLH NIHR BRC and North Thames NIHR ARC
(Applied Research Collaboration) and as a NIHR
Senior Investigator. KPR receives research support
from the National Institutes for Health. DF is funded
by the Wellcome Trust [205407/Z/16/Z].

**Competing interests:** The authors have declared
that no competing interests exist.

## Conclusions

Empathic compassion and understanding are distinct dimensions of empathy with differential demographic associations. Perspective taking may decline due to cognitive inflexibility with older age whereas empathic concern increases in older men suggesting it is socially-driven.

## Introduction

Empathy comprises the ability to feel compassion for another person's experience (emotional empathy) and the cognitive capacity to take the mental perspective of another person in order to understand their feelings (cognitive empathy) [1]. As a key component of social cognition [2], empathy is fundamental to guiding prosocial behaviour [3]. Understanding empathy is important as higher levels of empathy are associated with higher life satisfaction [4], and lower rates of loneliness [5] and depression [6], especially in carers and healthcare professionals. Emotional and cognitive empathy, measured by self-report using the Interpersonal Reactivity Index [7] domains of empathic concern (akin to emotional empathy) and perspective taking (cognitive empathy), are moderately correlated [7,8] but appear to be distinct dimensions of empathy with potentially different drivers and consequences [9]. Identifying the links between empathy and static characteristics such as gender and ethnicity or dynamic factors such as aging, education, employment, or social behaviours including marriage and social contact, has the potential to elucidate the determinants of empathy, such as learning through social interaction [10], and how empathy changes during the life-course [1].

Understanding how empathy changes with age and to what extent this is related to biological and psychosocial factors is important as it may clarify the mechanisms involved and links to neurodegenerative diseases in which loss of empathy can be characteristic [11], as well as identify targets to improve empathy. Several studies have suggested decline in empathy in older age [1], for example, a large nationally representative cross-sectional study of US adults, aged 18–90 years old, born between 1920 and 1999, found an inverse-U shaped association of empathy with age, so that both emotional and cognitive empathy increased from young adulthood and peaked in middle-age—around age 60—before declining [12].

However, there is uncertainty to what extent the association with age is a cohort effect [1], meaning that individuals may decline in empathy related to life experience or symptoms of cognitive decline, or observed age differences may be related to successive generations having differing early life experiences and education levels. A previous study suggesting a cohort effect but no longitudinal decline in empathy [13] did not differentiate between cognitive and emotional empathy which is likely to be important as cognitive empathy may be more susceptible to aging. If the reduction is because of cognitive decline then the ability to take the perspective of another person (or cognitive empathy), might be expected to be more strongly associated with age than empathic concern, as cognitive empathy is more impaired than empathic concern in patients with neurodegenerative disease [11], and the association will be there in cohorts from differing cultures.

Several sociodemographic characteristics have been reported to be associated with empathy. Women score higher on empathy scales than men [12,14,15] and personality traits, including conscientiousness, openness to experience, and agreeableness are linked to higher empathy [8,16,17]. However, studies of personality and empathy have usually been carried out in specific groups such as students or adolescents and the relationship may differ in other groups

[8,16]. Empathy would be expected to be related to career choice with those choosing the caring professions having higher emotional empathy, and higher levels of cognitive empathy are associated with better mental health outcomes for informal family carers [18] and healthcare professionals [19], suggesting that cognitive empathy may mediate stressful aspects of providing care to another person perhaps by allowing understanding of the feelings of the person cared for. However, empathy varies in different countries and cultures [17] so examination of empathy and its associations in specific settings is important. Furthermore, considering a range of different characteristics may elucidate the nature of the relationships of these factors with empathy. For example, factors which can be considered constant in individuals, such as ethnicity, gender or personality cannot be consequences but may be causes of empathy, whereas fluid characteristics like employment, socioeconomic status or choosing a partner may be the product, or also potentially cause, of empathy.

Therefore, in this study, we aim to describe levels of self-reported empathic concern and perspective taking for the first time in a large UK population, and according to a range of sociodemographic characteristics. We describe cross-sectional associations of empathic concern and perspective taking with age, gender, ethnicity, relationship status, education, occupation, caring responsibilities and personality. Our specific objectives are to test the following research hypotheses:

1. Older age will be more strongly negatively associated with perspective taking than empathic concern

2. Self-reported empathy will be associated with female gender, socio-economic and relationship status, education, and personality traits.

3. Empathic concern but not perspective taking will be associated with being a carer and working in caring professions

## Methods

### Study design and participants

This study is a cross-sectional analysis of data from the COVID-19 Social Study [20], a longitudinal cohort study of UK-dwelling participants aged 18 years and older. The COVID-19 Social Study started on 21st March 2020 to consider the psychological and social effects of the COVID-19 pandemic and the resulting restrictions on adults in the UK. Although the sample was not random, its large sample, which is well-stratified to include differing groups and well-phenotyped, make it a suitable dataset for exploring broader psychological and social factors beyond the pandemic itself. The study was promoted through three primary routes to encourage participation from diverse and under-represented groups. Firstly, convenience sampling including promotion through existing mailing lists and networks including large databases of UK adults who had consented to contact about research; secondly we conducted targeted recruitment of individuals from low-income backgrounds, with no or few educational qualifications, or unemployed via partnership with recruitment firms; and thirdly we promoted the study to vulnerable groups including older people and those with mental illness through third sector organisations. The study was approved by the UCL Research Ethics Committee [12467/005] and all participants gave informed consent. Full details of the study protocol are available at www.covidsocialstudy.org.

As the questions about empathy were included during only one of the weekly study questionnaires, our analysis was cross-sectional. Flow of participants is described in S1 Fig. Eligibility criteria for participants in this analysis were 1) UK resident aged ≥18 years, 2) completing

a baseline questionnaire on joining the study between study inception on 21[st] March 2020 and 20[th] June 2020, which was the last date of inclusion of the questions about empathy, 3) completing the empathy questionnaire between 13[th] and 20[th] June (week 13 of the study).

## Measures

**Empathic concern and perspective taking.** We measured empathic concern (EC) and perspective taking (PT) domains using questions from the Interpersonal reactivity index (IRI) [7], a 28 item scale answered on a 5-point Likert scale ranging from "Does not describe me well" to "Describes me very well". It has four subscales measuring different dimensions of empathy. We used the 14 questions (Table 1) assessing empathic concern and perspective taking. Scores for the two subscales were averaged giving mean scores for empathic concern and perspective taking ranging from 1–5. Higher scores denote higher empathy.

Factor analysis in 1161 US college students showed the EC and PT subscales have low-moderate correlation (r = 0.33) [7]. The IRI has good psychometric properties and has been validated in several large population studies [9]. The scales have high internal reliability and test-retest reliability [9]. Internal consistency for these subscales, measured by Cronbach's alpha, have been reported to be between 0.70 and 0.78 [7,21]. Cronbach's alpha in our sample was 0.80 for the EC subscale and 0.81 for the PT scale. Construct validity is demonstrated by the correlation of the PT subscale with measures of cognitive empathy, and the EC scale with emotional empathy measures [15]. The IRI is considered to measure 'trait-based' empathy, meaning the long-term, rather than situational, tendency to empathise with others, thus should not reflect empathy related to the COVID-19 pandemic.

**Covariates.** We chose to include information on sociodemographic, lifestyle and personality factors based on our a priori hypotheses and previous literature. The following covariates were derived from the baseline questionnaire: age in 10 year categories; gender (male, female, other/prefer not to say); ethnicity (White, Other); relationship status (never married, divorced/widowed, in a relationship but not cohabiting, co-habiting with partner/spouse); and living status (alone or with others). Usual social contact frequency was assessed by asking 'how

**Table 1. Empathic concern and perspective taking subscales from Interpersonal reactivity index.**

Empathic concern:
1. I often have tender, concerned feelings for people less fortunate than me.
2. Sometimes I don't feel very sorry for other people when they are having problems. (*)
3. When I see someone being taken advantage of, I feel kind of protective towards them.
4. Other people's misfortunes do not usually disturb me a great deal. (*)
5. When I see someone being treated unfairly, I sometimes don't feel very much pity for them. (*)
6. I would describe myself as a pretty soft-hearted person.
7. I am often quite touched by things that I see happen.

Perspective-taking:
1. I sometimes find it difficult to see things from the "other guy's" point of view. (*)
2. I try to look at everybody's side of a disagreement before I make a decision.
3. I sometimes try to understand my friends better by imagining how things look from their perspective.
4. If I'm sure I'm right about something, I don't waste much time listening to other people's arguments. (*)
5. I believe that there are two sides to every question and try to look at them both.
6. When I'm upset at someone, I usually try to "put myself in his shoes" for a while.
7. Before criticizing somebody, I try to imagine how I would feel if I were in their place.

Responses were given on a 5-item Likert scale with two anchors (A = Does not describe me well; E = Describes me very well). Questions were scored from 1 to 5 and questions marked with (*) were reverse-scored.

often do you usually meet up with people face-to-face socially, not for work' (<weekly, 1-2/ week, 3+ weekly).

We also derived information about employment (School/university/employment, or not working); household income (<£30,000, ≥£30,000); education status (Lower-secondary or below, higher-secondary, or graduate); caring responsibilities for relatives, friends, people with long-term conditions, or grandchildren (Y/N); and whether employed as a health or social care worker, teacher or childcare worker, or other keyworker role as defined by the UK government e.g. public service, utility or transport worker, none. Participants were also asked whether they had any pre-existing long-term physical illness (hypertension, diabetes, heart disease, lung disease, cancer, another chronic condition).

Personality was measured using the Big Five Inventory (BFI-2), which measures five domains derived from 15 facets: extraversion, agreeableness, conscientiousness, nervousness, and openness [22]. Respondents rated their agreement with each statement using a 7-point scale ranging from "strongly disagree" to "strongly agree", and we created scores ranging from 3–21 by summing together the three questions for each personality domain.

### Analyses

We first described the demographics of the sample, and physical health and personality traits. Variables are described using means and standard deviations (SD) for continuous variables and frequencies and percentages for categorical variables. We then examined empathic concern and perspective taking and reported these according to age and gender. We examined perspective taking and empathic concern in separate models as the domains have low-moderate correlation. We present unweighted results as well as results weighted to the proportions of age group, gender and educational level on the basis of Office for National Statistics (ONS) population estimates and Annual Population Survey [23], to account for the non-random nature of the sample. Our sample size was not predetermined but post-hoc power calculation indicated that 30,033 participants gave >99% power at 5% significance level to detect the moderate effect size (d = 0.50) found in previous comparisons of empathy by gender [12].

**Association of empathy with sociodemographic characteristics.**   s We assessed associations of empathic concern and perspective taking with age, gender, ethnicity, education level, living situation, marital status, employment, household income, keyworker status, carer status, usual face-to-face contact, physical health, and 'big-five' personality characteristics. We first used univariable linear regression to assess unadjusted associations and then included all variables in a multivariable linear regression model.

**Sensitivity analyses.**   We repeated the above analyses without weighting. In additional analyses, as 11% of the cohort had missing data on at least one predictor, we also conducted sensitivity analyses using multiple imputation by chained equations [24] for missing covariates to maximise statistical power. We used the *mi* package in STATA to create ten imputed datasets constructed from all potential covariate and outcome variables, before using linear regression on each imputed dataset with weighting as above, and used Rubin's rules to combine coefficients.

All analyses were conducted using STATA SE version 14.2 (Statacorp).

## Results

Analyses included 30,033 people and full characteristics of the sample are in Table 2. Three-quarters (22,461) of participants were women and the mean age was 54.4 (SD 14.1) years. There was a wide age range for female and male participants, with a higher proportion of male participants in the 65–74 (31.1%) and ≥75 (9.3%) age-groups, than for women (19.1 and 4.2%

**Table 2. Characteristics of the sample (n = 30,033).**

| Characteristic | Category | Unweighted N (%) * Mean (sd, range) | Weighted % * Mean (95% CI) |
|---|---|---|---|
| **Age (years)** | Mean (sd, range) | 54.4 (14.1, 18–90) * | 55.0 (54.7, 55.3) * |
| | 18–25 | 528 (1.8) | 4.4% |
| | 25–34 | 2,612 (8.7) | 8.6% |
| | 35–44 | 4,524 (15.1) | 12.4% |
| | 45–54 | 6,310 (21.0) | 17.1% |
| | 55–64 | 7,819 (26.0) | 25.2% |
| | 65–74 | 6,615 (22.0) | 25.6% |
| | ≥75 | 1,625 (5.4) | 6.7% |
| **Gender** | Female | 22,461 (74.8) | 50.8% |
| | Male | 7,455 (24.8) | 49.2% |
| | Other/prefer not to say | 117 (0.4) | |
| **Ethnicity** | White | 28,794 (96.1) | 91.7% |
| | Other | 1,442 (3.9) | 8.3% |
| | *Missing* | 97 | |
| **Educational level** | Lower secondary | 4,139 (13.8) | 32.1% |
| | Higher secondary | 5,072 (16.9) | 31.7% |
| | Graduate | 20,822 (69.3) | 36.2% |
| **Living situation** | Lives alone | 6,494 (21.6) | 21.3% |
| | Lives with others | 23,536 (78.4) | 78.7% |
| | *Missing* | 3 | |
| **Marital status** | Cohabiting with partner/spouse | 19,272 (64.2) | 62.2% |
| | Living apart from partner/spouse | 1,648 (5.5) | 5.9% |
| | Divorced/widowed | 4,567 (15.2) | 14.6% |
| | Single, never married | 4,546 (15.1) | 17.3% |
| **Employment** | In employment | 17,323 (57.7) | 50.2% |
| | Retired/not working | 12,710 (42.3) | 49.8% |
| **Household income** | < £30,000 | 10,937 (40.5) | 50.6% |
| | ≥ £30,000 | 16,093 (59.5) | 49.4% |
| | *Prefer not to say* | 3,003 | |
| **'Keyworker' status: employed in the following jobs:** | Health, social care or relevant related support worker | 2,807 (9.4) | 6.9% |
| | Teacher or childcare worker still travelling to work | 965 (3.2) | 2.1% |
| | Other 'keyworker' | 2,368 (7.9) | 9.2% |
| | None of these | 23,893 (79.6) | 81.7% |
| **Self-described carer** | | 4,618 (15.4) | 14.0% |
| **Usual face-to-face contact with others socially** | Less than once weekly | 8,497 (28.4) | 30.9% |
| | Once or twice per week | 10,183 (34.0) | 33.5% |
| | Three or more per week | 11,295 (37.7) | 35.5% |
| **Having a long-term health condition** | | 12,786 (42.6) | 46.8% |
| **Personality characteristics mean score (sd, range) (missing = 123)** | Neuroticism | 11.1 (4.3, 3–21) * | 11.0 (10.9, 11.1) * |
| | Extroversion | 12.8 (4.3, 3–21) * | 12.6 (12.5, 12.7) * |
| | Openness to experience | 15.4 (3.3, 3–21) * | 14.9 (14.9, 15.0) * |
| | Agreeableness | 15.5 (3.0, 3–21) * | 15.4 (15.3, 15.4) * |
| | Conscientiousness | 16.0 (2.9, 3–21) * | 15.8 (15.7, 15.8) * |

*(Continued)*

**Table 2.** (Continued)

| Characteristic | Category | Unweighted N (%) * Mean (sd, range) | Weighted % * Mean (95% CI) |
|---|---|---|---|
| Interpersonal reactivity index mean (sd, range) | Empathic concern | 3.97 (0.66, 1–5) * | 3.86 (3.85, 3.88) * |
| | Perspective taking | 3.67 (0.69, 1–5) * | 3.57 (3.56, 3.59) * |

sd = standard deviation. Weighted data matched to the UK proportions of gender, age, ethnicity, education and country of living from the Office for National Statistics (ONS, 2018).

respectively). Respondents were predominantly from white ethnic groups (96.1%) and 69.3% had attained a degree or higher education level. The majority (57.7%) were in employment and 59.5% reported a household income of £30,000 or greater. The majority (64.2%) were cohabiting with a partner or spouse and 21.6% lived alone. One-fifth of respondents reported working in a 'keyworker' role, including 9.4% who worked in health or social care.

Mean empathic concern score was 3.97 (SD 0.66, range 1–5) and mean perspective taking score was 3.67 (0.69, 1–5); 20% of the variance between these domains was shared (r = 0.45, p < 0.001). Mean scores weighted by gender, age, ethnicity, education, and country of residence within the UK were 3.86 for empathic concern (95% confidence interval 3.85, 3.88) and 3.57 for perspective taking (3.56, 3.59).

Mean scores for women were 4.06 for empathic concern (0.63, 1–5) and 3.74 for perspective taking (0.68, 1–5) and for men 3.70 for empathic concern (0.66, 1–5) and 3.48 for perspective taking (0.69, 1–5). Empathy scores varied for men and women by age. For women, there was an inverse-u association for both empathic concern and perspective taking. For men, empathic concern increased and perspective taking decreased with older age (Fig 1; Quadratic line of best fit due to non-linear associations (p<0.001)). Empathy scores according to all characteristics are in S1 Table.

### Association of sociodemographic characteristics with empathic concern

Complete data were obtained from 25,169 (83.8%) of participants, on whom we performed our primary complete case analyses.

In multivariable models, weighted for UK population distribution (Table 3), mean empathic concern score was 0.23 (0.21, 0.26) points higher for women than men, 0.08 (0.02, 0.14) points higher for non-white respondents than white respondents, 0.04 (0.00, 0.07) points higher for those with graduate v lower secondary education, 0.11 (0.06, 0.15) points higher for health or social care workers and 0.06 (0.00, 0.11) higher for teachers or childcare workers compared to those not in these or other 'keyworker' roles. Each standard deviation higher score on neuroticism was associated with 0.11 higher empathic concern score (0.10, 0.12). For one standard deviation higher extroversion score, empathic concern was 0.06 points higher (0.05, 0.07). Each standard deviation higher openness to experience was associated with 0.11 higher empathic concern (0.10, 0.13) and for each standard deviation of agreeableness, empathic concern was 0.24 points higher (0.23, 0.25) (Table 3).

Results were consistent in analyses without population weighting (S2 Table), and in models based on the full sample of 30,033 respondents with multiple imputation for missing data (S3 Table).

### Association of sociodemographic characteristics with perspective taking

In weighted multivariable models (Table 4), mean perspective taking score was associated with younger age (0.23 (0.34, 0.13) points higher for 18–25 year olds compared to respondents aged

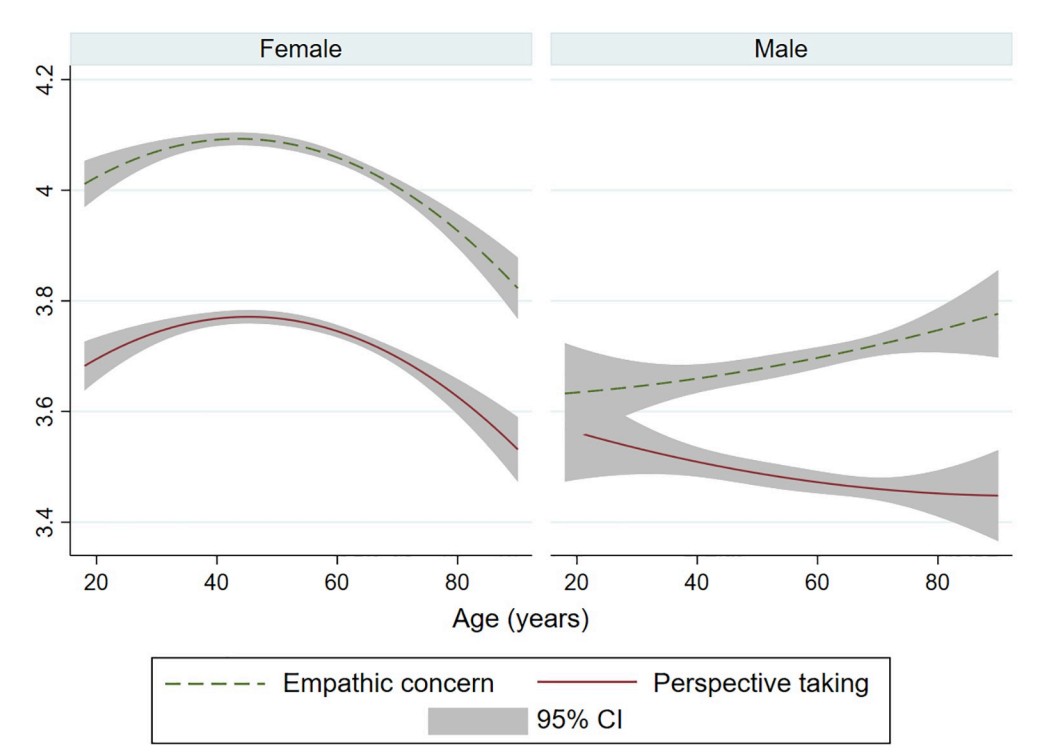

**Fig 1. Mean empathic concern and perspective taking scores by age and gender.** Predicted mean empathic concern (EC) and perspective taking (PT) by age in women and men, from regression of EC or PT on age and age$^2$ with 95% confidence interval (CI).

75 years or older). Mean score was 0.18 (0.16, 0.21) points higher for women and higher for those with more education (0.09 (0.06, 0.12) points higher for graduate v those with lower secondary education). Being a health or social care worker was associated with 0.05 (0.01, 0.10) higher score. Each standard deviation lower of neuroticism, and higher of openness to experience and agreeableness were associated with 0.08 (0.06, 0.08), 0.09 (0.08, 0.11), and 0.26 (0.24, 0.27) higher perspective taking scores respectively (Table 4).

Results were similar in analyses without weighting for population norms (S4 Table), and in models in which we imputed missing data on the full sample of 30,033 respondents (S3 Table).

## Discussion

This study is the first to describe self-reported empathic concern and perspective taking in a large UK sample. We found variations in empathy according to age and these associations also differed between women and men. There was an inverse-u association for empathic concern and perspective taking for women, with both improving with age and then beginning to fall from around age 45. Women's scores did not fall to the level of men's, even in older age up to the age of 90 years, when men's empathic concern increased, as their perspective taking decreased. In multivariable models, perspective taking, but not empathic concern, was independently associated with age, and both domains were higher in women than men and in those with more education. Those with health and social care jobs scored higher in empathic concern and perspective taking, and teachers and childcare workers had higher empathic concern scores, as did non-white people. Higher neuroticism was associated with more empathic

**Table 3. Association of participant characteristics with empathic concern (n = 25,169).**

| | | Unweighted univariable Complete cases | | Weighted multivariable | |
|---|---|---|---|---|---|
| | | Coefficient | P value | Coefficient | P value |
| **Age (years)** | 18–25 | Reference | <0.001 | Reference | 0.04 |
| | 25–34 | -0.07 (-0.18, 0.03) | | -0.06 (-0.15, 0.03) | |
| | 35–44 | -0.12 (-0.23, -0.02) | | -0.06 (-0.15, 0.03) | |
| | 45–54 | -0.10 (-0.21, -0.00) | | -0.02 (-0.10, 0.07) | |
| | 55–64 | -0.14 (-0.24, -0.04) | | -0.01 (-0.10, 0.08) | |
| | 65–74 | -0.16 (-0.26, -0.06) | | 0.02 (-0.07, 0.11) | |
| | ≥75 | -0.21 (-0.32, -0.09) | | 0.01 (-0.10, 0.11) | |
| **Gender** | Male | Reference | <0.001 | Reference | <0.001 |
| | Female | 0.37 (0.34, 0.39) | | 0.23 (0.21, 0.26) | |
| **Ethnicity** | White | Reference | 0.18 | Reference | 0.01 |
| | Other | 0.05 (-0.02, 0.12) | | 0.08 (0.02, 0.14) | |
| **Educational level** | Lower secondary (ref) | Reference | <0.001 | Reference | 0.02 |
| | Higher secondary | 0.04 (0.00, 0.08) | | 0.00 (-0.03, 0.04) | |
| | Graduate | 0.09 (0.06, 0.12) | | 0.04 (0.00, 0.07) | |
| **Living status** | Alone (ref) | Reference | 0.20 | Reference | 0.92 |
| | With others | 0.02 (-0.01, 0.05) | | 0.00 (-0.04, 0.04) | |
| **Marital status** | Single (ref) | Reference | 0.004 | Reference | 0.14 |
| | Divorced/widowed | 0.09 (0.04, 0.14) | | 0.02 (-0.02, 0.07) | |
| | Non cohabiting partner | 0.06 (-0.01, 0.14) | | 0.04 (-0.02, 0.10) | |
| | Married/cohabiting | 0.03 (-0.01, 0.07) | | 0.05 (0.01, 0.10) | |
| **Employment** | Not working (ref) | Reference | 0.11 | Reference | 0.86 |
| | Working | 0.02 (-0.01, 0.05) | | 0.00 (-0.03, 0.03) | |
| **Household income** | < £30,000 (ref) | Reference | 0.54 | Reference | 0.66 |
| | ≥ £30,000 | -0.01 (-0.04, 0.02) | | -0.01 (-0.03, 0.02) | |
| **'Keyworker' status** | None of these (ref) | Reference | <0.001 | Reference | <0.001 |
| | Health/social-care | 0.17 (0.12, 0.22) | | 0.11 (0.06, 0.15) | |
| | Teacher/childcare | 0.22 (0.15, 0.29) | | 0.06 (0.00, 0.11) | |
| | Other 'keyworker' | -0.08 (-0.13, -0.03) | | -0.01 (-0.05, 0.04) | |
| **Carer status** | Not carer (ref) | Reference | <0.001 | Reference | 0.57 |
| | carer | 0.09 (0.05, 0.12) | | 0.01 (-0.02, 0.04) | |
| **Face-to-face social contact** | < 1 time per week (ref) | Reference | <0.001 | Reference | 0.10 |
| | 1–2 times per week | 0.07 (0.03, 0.10) | | 0.02 (-0.01, 0.04) | |
| | 3+ times per week | 0.13 (0.09, 0.16) | | 0.03 (0.07, 0.00) | |
| **Long-term condition** | No (ref) | Reference | 0.88 | Reference | 0.12 |
| | Yes | 0.00 (-0.03, 0.03) | | 0.02 (-0.01, 0.04) | |
| **Personality mean score (per one standard deviation higher)** | Neuroticism | 0.09 (0.07, 0.10) | <0.001 | 0.11 (0.10, 0.12) | <0.001 |
| | Extroversion | 0.11 (0.10, 0.12) | <0.001 | 0.06 (0.05, 0.07) | <0.001 |
| | Openness to experience | 0.14 (0.13, 0.16) | <0.001 | 0.11 (0.10, 0.13) | <0.001 |
| | Agreeableness | 0.27 (0.25, 0.28) | <0.001 | 0.24 (0.23, 0.25) | <0.001 |
| | Conscientiousness | 0.10 (0.08, 0.11) | <0.001 | -0.01 (-0.02, 0.01) | 0.48 |

Linear regression models weighted to the UK proportions of gender, age, ethnicity, education and country of living obtained from the Office for National Statistics (ONS, 2018). Multivariable models are mutually adjusted for included variables. Coefficients indicate estimated difference in interpersonal reactivity index empathic concern score according to respondent characteristic.

**Table 4. Association of participant characteristics with perspective taking (n = 25,169).**

| | | Weighted univariable | | Weighted multivariable | |
|---|---|---|---|---|---|
| | | Coefficient | P value | Coefficient | P value |
| **Age (years)** | 18–25 | Reference | <0.001 | Reference | <0.001 |
| | 25–34 | -0.01 (-0.09, 0.12) | | -0.05 (-0.14, 0.04) | |
| | 35–44 | -0.00 (-0.11, 0.10) | | -0.04 (-0.13, 0.04) | |
| | 45–54 | -0.05 (-0.15, 0.05) | | -0.09 (-0.18, -0.01) | |
| | 55–64 | -0.11 (-0.21, -0.01) | | -0.13 (-0.22, -0.04) | |
| | 65–74 | -0.12 (-0.22, -0.02) | | -0.13 (-0.22, -0.04) | |
| | ≥75 | -0.24 (-0.36, -0.12) | | -0.23 (-0.34, -0.13) | |
| **Gender** | Male | Reference | <0.001 | Reference | <0.001 |
| | Female | 0.27 (0.24, 0.29) | | 0.18 (0.16, 0.21) | |
| **Ethnicity** | White | Reference | 0.68 | Reference | 0.99 |
| | Other | 0.02 (-0.06, 0.09) | | -0.00 (-0.06, 0.06) | |
| **Educational level** | Lower secondary (ref) | Reference | <0.001 | Reference | <0.001 |
| | Higher secondary | 0.12 (0.08, 0.16) | | 0.06 (0.02, 0.10) | |
| | Graduate | 0.18 (0.15, 0.22) | | 0.09 (0.06, 0.12) | |
| **Living status** | Alone (ref) | Reference | 0.56 | Reference | 0.57 |
| | With others | 0.01 (-0.02, 0.04) | | -0.01 (-0.06, 0.03) | |
| **Marital status** | Single (ref) | Reference | 0.02 | Reference | 0.39 |
| | Divorced/widowed | 0.06 (0.01, 0.11) | | 0.04 (-0.01, 0.09) | |
| | Non cohabiting partner | 0.04 (-0.05, 0.12) | | 0.03 (-0.04, 0.10) | |
| | Married/cohabiting | -0.00 (-0.04, 0.04) | | 0.02 (-0.02, 0.07) | |
| **Employment** | Not working (ref) | Reference | <0.001 | Reference | 0.73 |
| | Working | 0.08 (0.05, 0.11) | | 0.01 (-0.03, 0.04) | |
| **Household income** | < £30,000 (ref) | Reference | <0.001 | Reference | 0.25 |
| | ≥ £30,000 | 0.04 (0.02, 0.05) | | 0.02 (-0.01, 0.05) | |
| **'Keyworker' status** | None of these (ref) | Reference | <0.001 | Reference | 0.07 |
| | Health/social-care | 0.17 (0.13, 0.21) | | 0.05 (0.01, 0.10) | |
| | Teacher/childcare | 0.17 (0.09, 0.26) | | -0.02 (-0.09, 0.06) | |
| | Other 'keyworker' | -0.01 (-0.07, 0.02) | | 0.01 (-0.03, 0.06) | |
| **Carer status** | Not carer (ref) | Reference | <0.001 | Reference | 0.09 |
| | carer | 0.08 (0.05, 0.12) | | 0.03 (-0.00, 0.06) | |
| **Face-to-face social contact** | < 1 time per week (ref) | Reference | <0.001 | Reference | 0.90 |
| | 1–2 times per week | 0.06 (0.02, 0.09) | | 0.01 (-0.02, 0.04) | |
| | 3+ times per week | 0.09 (0.05, 0.12) | | 0.00 (-0.03, 0.03) | |
| **Long-term condition** | No (ref) | Reference | <0.001 | Reference | 0.99 |
| | Yes | -0.07 (-0.10, -0.04) | | -0.00 (-0.03, 0.03) | |
| **Personality mean score (per one standard deviation higher)** | Neuroticism | -0.06 (-0.08, -0.05) | <0.001 | -0.08 (-0.08, -0.06) | <0.001 |
| | Extroversion | 0.07 (0.06, 0.08) | <0.001 | -0.01 (-0.02, 0.01) | 0.22 |
| | Openness to experience | 0.13 (0.11, 0.14) | <0.001 | 0.09 (0.08, 0.11) | <0.001 |
| | Agreeableness | 0.29 (0.27, 0.30) | <0.001 | 0.26 (0.24, 0.27) | <0.001 |
| | Conscientiousness | 0.12 (0.11, 0.14) | <0.001 | 0.01 (-0.01, 0.02) | 0.20 |

Linear regression models weighted to the UK proportions of gender, age, ethnicity, education and country of living obtained from the Office for National Statistics (ONS, 2018). Multivariable models are mutually adjusted for included variables. Coefficients indicate estimated difference in interpersonal reactivity index perspective taking score according to respondent characteristic.

concern and less perspective taking, extroversion was associated with higher empathic concern, and openness to experience and agreeableness were associated with higher scores on both subscales.

The mean scores in this study are similar to those in the initial 1980 validation paper which examined empathy using the interpersonal reactivity index in US college students [7]. It found mean empathic concern in women 4.10 and men 3.72, compared to 4.06 and 3.70 respectively in our sample, and perspective taking to be 3.57 in women and 3.40 in men, compared to 3.74 and 3.48 in our study. Differences in self-reported empathy may be partly attributable to variation across different countries, for example the lowest scores of 3.15 and 3.16 on EC and PT respectively were in Lithuania, and the highest scores in Ecuador (4.12 and 3.82), meaning that comparison to different countries may be invalid. UK data from an international study had mean empathic concern 3.49 and perspective taking 3.44 in 2,754 people aged 37 years on average [17], which was lower than in our study; and another UK study of first year medical students found women to have mean EC and PT 4.01 and 3.77 and men 3.78 and 3.57, which was more similar to our results [25]. Our study is based on a larger sample, with better coverage of age and social groups and is the best stratified assessment of empathy to date, so should provide the most accurate normative figures.

We expected that perspective taking would reduce with greater age, but that empathic concern may not. This hypothesis was due to the cognitive requirements of taking another person's perspective, which is reflected in evolutionary models of empathy whereby perspective taking is a higher-level process than experiencing empathic concern, and thus more susceptible to loss through cognitive dysfunction [26]. Our results were overall supportive of this hypothesis, with higher perspective taking associated with younger age in multivariable analyses, but no association between empathic concern and age in these adjusted models. There is evidence of age-related deficits in the cognitive ability to accurately perceive another person's emotions [27] and deficits in empathy are associated with structural abnormalities in the dorsal medial prefrontal cortex [28] and insula [29] as well as right temporoparietal deficits in healthy older people and those with neurodegenerative diseases [30]. Furthermore, cognitive empathy was shown in a recent meta-analysis to be associated with cognitive flexibility, whereas emotional empathy [31] was not. This lends weight to our hypothesis that decline in the cognitive ability to empathise with others may be linked with age-related changes in neural substrate, although other possibilities related to cultural and environmental changes in cognitive empathy are also possible.

The inverse relationship between age and perspective taking or cognitive empathy has been found in several studies [32–34], as has the lack of association of emotional empathy with age [32,33]. One study of 400 people with ages ranging from 10 to 87 years examined empathy both cross-sectionally across the age-range and longitudinally, with 260 (65%) participants participating in the 6 year assessment and 171 (43%) completed a 12 year follow-up, using an assessment combining emotional and cognitive empathy. Empathy was inversely associated with age in the cross-sectional analysis but there was no evidence of longitudinal decline [13], which may suggest that age associations of empathy are a cohort effect rather than an indication of age-related decline. However as our study shows, there may be differential associations of cognitive and emotional empathy so combining these may obscure change, and the sample size and follow-up time may have been insufficient in this study to detect change.

Our study builds on the existing evidence as our stratified analysis of empathic concern and perspective taking by gender indicated that the differences in the association of these two domains with age was partly due to the higher self-reported empathic concern in older men. This is, to our knowledge, the first finding of this strong age-gender effect on empathic

concern which persisted independent of potential confounders. Future studies should consider in greater detail what drives the effect of gender on empathy's association with age.

Several potential explanations for the association of age with empathy have been previously proposed [1]. A US study of 60 people reported an association between age and empathy which became non-significant when adjusted for level of education, suggesting that the age difference reflected different levels of education [35]. Another study of 1,567 people from a 1985 community sample of 22–92 year olds also reported the age-empathy association to be moderated by education [36]. However, in our larger study the association of age persisted after adjustment for educational level.

Women's higher empathy ratings compared to men found in this study are consistently shown in observational studies which use either self-rating scales, implicit measurements, or task-based assessment of empathy [14]. Our study adds that this persists independently of potential confounding effects of personality factors and social contact frequency. There are several potential explanations for women having higher empathy than men. Firstly, there may be culturally-driven differences in socialisation behaviour by gender [37] through social learning [10]. Secondly, differences may be explained through biological mechanisms, such as hormone-induced alterations in brain development through lower testosterone levels [38] or differential response to oxytocin [39], with genetic selection favouring more empathic women. Finally, social desirability bias may account for gender differences in self-reported empathy, supported by discrepancies in self-reported and task-based assessment of empathy where women over-reported their own empathy [40].

The strongest associations with empathy found in our study were for personality traits, in particular, agreeableness and openness to experience which had strong positive associations with both empathy subscales, and neuroticism which had a positive independent association with empathic concern and a negative association with perspective taking. Previous studies have found personality traits to account for around 20% of the variance of these domains [8], although studies have often been of students who may differ from a more general population [41]. One international study examined correlations between the big five personality characteristics and found similar inverse correlation between neuroticism and perspective taking, and, in age and sex-adjusted models, that empathic concern was most closely associated with agreeableness in 896 participants with mean age 21 years in China, Denmark, Germany and the United States [41]. Our study adds that these associations are independent of other potentially important confounders such as education, and seen throughout the lifespan. The strong association between personality and empathy may be partly due to response bias for both of these self-reported domains whereby individuals may have bias towards uniformly reporting that all of their personality and empathy traits are high, leading to inflation of the association.

Our findings of increased perspective taking and empathic concern in healthcare workers is consistent with our hypothesis. Some previous research suggests that empathy improves during healthcare training [42], consistent with the Learned Matching hypothesis, whereby empathy emerges and strengthens during development through domain-general processes of associative learning [10]. However other studies have suggested that empathy is stable during training [25], and having a higher level of empathy may mean that a person is more likely to choose such roles. Associations between professional caring roles (such as health/social-care work and childcare/teaching) and empathy were stronger for empathic concern than for perspective taking. For perspective taking, no association was found for teachers, or for informal caring (e.g. for a friend, relative or grandchild). Higher levels of empathy in healthcare professionals have been associated with lower rates of burnout [43] suggesting that these social cognitive domains are critical buffers to the potentially detrimental effects of the stress of a caring role.

We found that more education was associated with both empathy domains, more strongly with perspective taking than empathic concern, which is consistent with some previous studies [35]. Our study adds that the association persisted after adjustment for potential explanatory factors such as age and income. Having more education may increase cognitive ability to consider others' perspective and provide skills which encourage thoughts about others [36]. Empathic concern, but not perspective taking, was higher in non-White respondents, possibly reflecting the cultural differences in empathy demonstrated in large international studies [17,41], but the small number of non-White participants meant that we could not examine this in more detail considering different ethnic groups. The other socio-demographic factors we examined were not independently associated with empathy, including living alone, marital status, and social contact with others, being in employment or having high income, and having long-term health conditions. These factors are largely fluid and can vary throughout adulthood, unlike gender, education and personality, and their lack of association suggests that empathy may be largely determined by gender, personality and education, with subsequent changes over time related to aging.

## Strengths and limitations

This study's strengths include its large sample size and extensive measurement of sociodemographic and lifestyle factors. However, there are limitations. Though the study is not nationally-representative, the sample has good stratification across all major socio-demographic groups and our analyses were weighted to population estimates. Whilst the recruitment strategy deliberately over-sampled from disadvantaged groups, more extreme experiences may not be adequately captured. Less than 4% of the population were from minority ethnic backgrounds and we had relatively fewer responses from people at the extremes of age, particularly adult men aged under 25, meaning that our findings in these groups are less certain.

Empathy was measured in the 13[th] week of the study, meaning that empathy may have been measured up to 3 months apart from covariate measurement. In addition, this study was conducted during the first wave of COVID-19 pandemic, during 'lockdown' in the UK at a time of significant stress. However, the Interpersonal Reactivity Index is conceptually designed to measure long-term empathic traits, rather than empathy in specific personal or wider social situations, these factors are unlikely to affect the associations we found.

Our study is a cross-sectional analysis making it difficult to be certain of the mechanism and direction of the associations we found and we focused on sociodemographic associations, rather than the impact of empathy on mental health. Our self-report assessment of empathy potentially is susceptible to social desirability bias [44] though observer ratings correlate moderately with self-report [45] and links between empathy self-report and social behaviour support the scale's validity [15,17]. Finally, we did not have data on participants' general cognition and, although our study procedures means it is unlikely that participants had substantial cognitive impairment, it is possible that cognitive impairment in some participants would cause measurement error.

## Implications

This is the first large-scale study of empathy in a UK population, providing normative data, and we report similar levels of self-reported empathic concern and perspective taking in this population to international comparators. We describe the associations of empathy with a range of key demographic, lifestyle and personality characteristics. We found that perspective taking declines with greater age but that empathic concern does not, partly driven by higher levels of empathic concern in older men, lending weight to theories that declining perspective

taking partly reflects reduced cognitive flexibility with older age, while empathic concern may reflect social and cultural influences. Additionally, empathy may be biologically and culturally determined by gender, education and personality early in life, but unaffected by other social factors determined in adulthood, with the exception of vocation, where empathy may guide choice of career.

Future research should seek to clarify the mechanisms for the associations we have identified in our epidemiological analysis through further detailed examination of relevant factors. In particular, it is important to understand whether the decline in perspective taking is general or specific to those who will go on to develop cognitive impairment, and which biological and psycho-social mechanisms drive gender differences in age-related empathy changes. Self-reported empathy is higher in people working in caring professions independent of personality and educational level and, considering the key importance of this domain in guiding social behaviour and protecting against adverse mental health outcomes, future studies should determine the extent to which this can be modified or is an inherent trait which should be sought for these roles. As empathy is a crucial building block of social interactions, understanding more about greater empathy is crucially important to societal function.

## Supporting information

**S1 Fig. Flow of participants.** Empathy questions were included in the study during week 13-20th June 2020.
(TIF)

**S1 Table. Mean unweighted scores on empathic concern and perspective taking scales according to sociodemographic characteristics (n = 25,169).** sd = standard deviation; acategorised into tertiles based on distribution in this sample.
(DOCX)

**S2 Table. Association of participant characteristics with empathic concern–unweighted univariable and multivariable associations (n = 25,169).** Linear regression models. Multivariable models are mutually adjusted for included variables. Coefficients indicate estimated difference in Interpersonal reactivity index empathic concern score according to respondent characteristic.
(DOCX)

**S3 Table. Multivariable weighted associations of participant characteristics with empathic concern or perspective taking with missing data imputed using multiple imputation (n = 30,033).** Linear regression models, mutually adjusted for included variables, and weighted to the UK proportions of gender, age, ethnicity, education and country of living obtained from the Office for National Statistics. Coefficients indicate estimated difference in Interpersonal reactivity index empathic concern or perspective taking score according to respondent characteristic.
(DOCX)

**S4 Table. Association of participant characteristics with perspective taking–unweighted univariable and multivariable associations (n = 25,169).** Linear regression models. Multivariable models are mutually adjusted for included variables. Coefficients indicate estimated difference in Interpersonal reactivity index perspective taking score according to respondent characteristic.
(DOCX)

## Acknowledgments

The researchers are grateful for the support of several organisations with their recruitment efforts including: the UKRI Mental Health Networks, Find Out Now, UCL BioResource, SEO Works, FieldworkHub, and Optimal Workshop. The study was also supported by HealthWise Wales, the Health and Care Research Wales initiative, which is led by Cardiff University in collaboration with SAIL, Swansea University.

## Author Contributions

**Conceptualization:** Andrew Sommerlad, Jonathan Huntley, Gill Livingston, Katherine P. Rankin, Daisy Fancourt.

**Data curation:** Andrew Sommerlad.

**Formal analysis:** Andrew Sommerlad.

**Funding acquisition:** Daisy Fancourt.

**Investigation:** Andrew Sommerlad, Jonathan Huntley.

**Methodology:** Andrew Sommerlad, Daisy Fancourt.

**Visualization:** Andrew Sommerlad.

**Writing – original draft:** Andrew Sommerlad, Jonathan Huntley, Gill Livingston, Katherine P. Rankin, Daisy Fancourt.

**Writing – review & editing:** Andrew Sommerlad, Jonathan Huntley, Gill Livingston, Katherine P. Rankin, Daisy Fancourt.

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
