## [Decision Letter · Decision Letter 0]

14 Jun 2021

PONE-D-21-14791

Empathy and its associations with age and sociodemographic characteristics in a large UK population sample

PLOS ONE

Dear Authors,

Thank you for submitting your manuscript to PLOS ONE. After careful consideration, we feel that it has merit but does not fully meet PLOS ONE’s publication criteria as it currently stands. Therefore, we invite you to submit a revised version of the manuscript that addresses the points raised during the review process.

We look forward to receiving your revised manuscript.

Kind regards,

Marcel Pikhart

Academic Editor

PLOS ONE

Journal Requirements:

Reviewers' comments:

Reviewer's Responses to Questions

**Comments to the Author**

1. Is the manuscript technically sound, and do the data support the conclusions?

Reviewer #1: Yes

Reviewer #2: Yes

Reviewer #3: No

2. Has the statistical analysis been performed appropriately and rigorously? 

Reviewer #1: Yes

Reviewer #2: Yes

Reviewer #3: No

3. Have the authors made all data underlying the findings in their manuscript fully available?

Reviewer #1: Yes

Reviewer #2: Yes

Reviewer #3: No

4. Is the manuscript presented in an intelligible fashion and written in standard English?

Reviewer #1: Yes

Reviewer #2: Yes

Reviewer #3: Yes

5. Review Comments to the Author

Reviewer #1: In my opinion, authors clearly stated the research questions, sufficiently explained the statistical methods used and relevant results.

Reviewer #2: It seems to me an interesting and little explored topic, as the authors say, however I have some comments in relation to the manuscript:

Introduction

On line 61 it says: “Identifying the links between empathy and static characteristics such as gender and ethnicity or dynamic factors such as aging, education, employment or social behaviors, including marriage and social contact, has the potential to improve our understanding of what determines empathy and how it changes during the course of life ”. The claim is without a citation.

Methodology

In the methodology it was mentioned that the study is longitudinal, although a cross-sectional analysis was made for the paper. It is important to go into more detail about how the sample was selected and present it in the flow chart. Furthermore, it is worth saying why, if the data are longitudinal, a cross-sectional analysis was chosen.

In methodology include: Variables are described using means (standard deviation [SD]) for continuous variables and frequencies and percentages for categorical variables, for example.

Although the data is said to come from the COVID-19 Social Study and the link is presented for more information about it, it is necessary for the methodology to clarify the central objective of the general study.

It seems appropriate to explain how people were surveyed. Is it representative of the entire population 18 years of age or older? (although something is said in the limitations, but not in the methodology).

Are the scales used to measure EC and PT previously validated in the study population?

Did they use cognitive impairment as an exclusion criterion? Can this be considered a bias of the study?

Reviewer #3: The analysis is not advanced, and deals with a deep topic shallowly

The researcher(s) should refer to modern theories about empathy

Demographic factors are insufficient to describe a social psychology phenomenon

6. PLOS authors have the option to publish the peer review history of their article (what does this mean?). If published, this will include your full peer review and any attached files.

Reviewer #1: No

Reviewer #2: **Yes: **Marcela Agudelo-Botero

Reviewer #3: No

---

## [Author Response · Author response to Decision Letter 0]

9 Aug 2021

Thank you for the comments of editors and reviewers. We have addressed these point by point below with our response in bold font and revised sections of the manuscript in italics. 

Editors comments

We have amended the manuscript to meet PLOS ONE’s style requirements and naming conventions.

2. We note that you have indicated that data from this study are available upon request. PLOS only allows data to be available upon request if there are legal or ethical restrictions on sharing data publicly. For more information on unacceptable data access restrictions, please see http://journals.plos.org/plosone/s/data-availability#loc-unacceptable-data-access-restrictions .

We have amended the data availability statement in the manuscript as below. Funding restrictions currently do not permit open publication of the data but they can be accessed via contact with the authors and will be made publicly available early next year. Our recent paper published in PLOS One had similar data restrictions (https://journals.plos.org/plosone/article?id=10.1371/journal.pone.0248919). 

The data files supporting the findings of this study are available on request from the study data access committee through data sharing agreement via f.bu@ucl.ac.uk. The full data is not currently publicly available due to funding arrangements but will be made publicly available following completion of the COVID-19 Social Study at the start of 2022.

We have explained the limitations on data availability in our cover letter.

Reviewer #1

In my opinion, authors clearly stated the research questions, sufficiently explained the statistical methods used and relevant results.

Thank you for your review of our paper and we are pleased that you thought that the study was clear and appropriately conducted.

Reviewer #2

It seems to me an interesting and little explored topic, as the authors say, however I have some comments in relation to the manuscript:

Thank you very much for your review and helpful comments which we think have improved our manuscript.

Introduction

On line 61 it says: “Identifying the links between empathy and static characteristics such as gender and ethnicity or dynamic factors such as aging, education, employment or social behaviors, including marriage and social contact, has the potential to improve our understanding of what determines empathy and how it changes during the course of life ”. The claim is without a citation.

We have added citations to support this statement regarding empathy as a potentially learned social behaviour or determined by aging (p4, lines 62-65):

Identifying the links between empathy and static characteristics such as gender and ethnicity or dynamic factors such as aging, education, employment, or social behaviours including marriage and social contact, has the potential to elucidate the determinants of empathy, such as learning through social interaction (1), and how empathy changes during the life-course (2).

Methodology

In the methodology it was mentioned that the study is longitudinal, although a cross-sectional analysis was made for the paper. It is important to go into more detail about how the sample was selected and present it in the flow chart. Furthermore, it is worth saying why, if the data are longitudinal, a cross-sectional analysis was chosen.

Thank you for highlighting this. The COVID-19 Social study is a longitudinal study, with weekly surveys from March 2020.However, the IRI empathy questions on which we base our analysis were included at one time point of the study and so our presented analysis is cross-sectional. We have clarified this in the methods section (p7, lines 126-127):

As the questions about empathy were included during only one of the weekly study questionnaires, our analysis was cross-sectional.

We have also added a supplementary figure describing participant flow:

S1 Figure. Flow of participants.

Notes: Empathy questions were included in the study during week 13-20th June 2020

In methodology include: Variables are described using means (standard deviation [SD]) for continuous variables and frequencies and percentages for categorical variables, for example.

We have added this to added this to the analysis section of the manuscript (p9, line 177-179):

Variables are described using means and standard deviations (SD) for continuous variables and frequencies and percentages for categorical variables.

Although the data is said to come from the COVID-19 Social Study and the link is presented for more information about it, it is necessary for the methodology to clarify the central objective of the general study.

We have explained the overall aims of the main study (p6, lines 113-115):

The COVID-19 Social Study started on 21st March 2020 to consider the psychological and social effects of the COVID-19 pandemic and the resulting restrictions on adults in the UK.

It seems appropriate to explain how people were surveyed. Is it representative of the entire population 18 years of age or older? (although something is said in the limitations, but not in the methodology).

Thank you, we have added detail about the approach to recruitment for this study (p6, lines 115-123):

Although the sample was not random, its large sample, which is well-stratified to include differing groups and well-phenotyped, make it a suitable dataset for exploring broader psychological and social factors beyond the pandemic itself. The study was promoted through three primary routes to encourage participation from diverse and under-represented groups. Firstly, convenience sampling including promotion through existing mailing lists and networks including large databases of UK adults who had consented to contact about research; secondly we conducted targeted recruitment of individuals from low-income backgrounds, with no or few educational qualifications, or unemployed via partnership with recruitment firms; and thirdly we promoted the study to vulnerable groups including older people and those with mental illness through third sector organisations.

Are the scales used to measure EC and PT previously validated in the study population?

The scale has not been specifically validated in this study population – a novel contribution of our study is that this is the first large-scale use of the Interpersonal Reactivity Index in the UK. However, the IRI has well-established psychometric properties from population-based studies in several countries and we have summarised this information in the manuscript (p8, lines 147-152):

The IRI has good psychometric properties and has been validated in several large population studies (3). The scales have high internal reliability and test-retest reliability (3). Internal consistency for these subscales, measured by Cronbach’s alpha, have been reported to be between 0.70 and 0.78 (4, 5). Cronbach’s alpha in our sample was 0.80 for the EC subscale and 0.81 for the PT scale. Construct validity is demonstrated by the correlation of the PT subscale with measures of cognitive empathy, and the EC scale with emotional empathy measures (6).

Did they use cognitive impairment as an exclusion criterion? Can this be considered a bias of the study?

We did not obtain data on the general cognition of study participants so were not able to use cognitive impairment as an exclusion criterion. Our approach recruitment and the online self-completion of questionnaires makes it unlikely that we had many participants with substantial cognitive impairment. However, we have added that there is the potential for measurement error through the inclusion of people with cognitive impairment (p24, lines 407-409):

Finally, we did not have data on participants’ general cognition and, although our study procedures means it is unlikely that participants had substantial cognitive impairment, it is possible that cognitive impairment in some participants would cause measurement error.

Reviewer #3

The analysis is not advanced, and deals with a deep topic shallowly. The researcher(s) should refer to modern theories about empathy

We have added to our manuscript to place our study more clearly within modern theories about empathy. In the introduction, we have clarified the value of considering socio-demographic determinants of empathy (p4, lines 62-65) as markers of social learning:

Identifying the links between empathy and static characteristics such as gender and ethnicity or dynamic factors such as aging, education, employment, or social behaviours including marriage and social contact, has the potential to elucidate the determinants of empathy, such as learning through social interaction (1), and how empathy changes during the life-course (2).

We have added to our discussion in several places to place our research in the context of the extant literature.

p20, lines 301-304: We expected that perspective taking would reduce with greater age, but that empathic concern may not. This hypothesis was due to the cognitive requirements of taking another person’s perspective, which is reflected in evolutionary models of empathy whereby perspective taking is a higher-level process than experiencing empathic concern, and thus more susceptible to loss through cognitive dysfunction (7).

p21, lines 342-349: There are several potential explanations of woman having higher empathy than men. Firstly, there may be culturally-driven differences in socialisation behaviour by gender (8) through social learning (1). Secondly, differences may be explained through biological mechanisms, such as hormone-induced alterations in brain development through lower testosterone levels (9) or differential response to oxytocin (10), with genetic selection favouring more empathic women. Finally, social desirability bias may account for gender differences in self-reported empathy, supported by discrepancies in self-reported and task-based assessment of empathy where women over-reported their own empathy (11).

P22, lines 367-369: Some previous research suggests that empathy improves during healthcare training (12), consistent with the Learned Matching hypothesis, whereby empathy emerges and strengthens during development through domain-general processes of associative learning (1).

Demographic factors are insufficient to describe a social psychology phenomenon

We agree that socio-demographic factors cannot fully explain complex social psychological phenomena and our study was not designed to perform a thorough investigation of how these demographic characteristics may interact with empathy. Instead, our aim was to perform an initial step of that investigation, by identifying large-scale epidemiological evidence of associations. Our study contributes to the understanding of how empathy may change during the life-course and how it differs according to socio-demographic characteristics, such as our novel finding of the gender difference in the associations of empathic concern and perspective taking by age. We therefore build upon previous research through the large UK based sample, our wide range of examined characteristics, and our consideration of potential confounders. Even though investigating detailed mechanisms of relationships between social factors and empathy was beyond the scope of this paper, this is an excellent suggestion for future lines of research and we have added that suggestion to the conclusion of our manuscript (p25, lines 434-442):

Future research should seek to clarify the mechanisms for the associations we have identified in our epidemiological analysis through further detailed examination of relevant factors. In particular, it is important to understand whether the decline in perspective taking is general or specific to those who will go on to develop cognitive impairment, and which biological and psycho-social mechanisms drive gender differences in age-related empathy changes. Self-reported empathy is higher in people working in caring professions independent of personality and educational level and, considering the key importance of this domain in guiding social behaviour and protecting against adverse mental health outcomes, future studies should determine the extent to which this can be modified or is an inherent trait which should be sought for these roles.

 

References

1. Heyes C. Empathy is not in our genes. Neuroscience & Biobehavioral Reviews. 2018;95:499-507.

2. Beadle JN, De la Vega CE. Impact of aging on empathy: Review of psychological and neural mechanisms. Frontiers in psychiatry. 2019;10:331.

3. Konrath S. A critical analysis of the Interpersonal Reactivity Index. MedEdPORTAL Directory and Repository of Educational Assessment Measures (DREAM). 2013.

4. Davis MH. A multidimensional approach to individual differences in empathy. JSAS Catalog of Selected Documents in Psychology. 1980;10:85.

5. Hemmerdinger JM, Stoddart SD, Lilford RJ. A systematic review of tests of empathy in medicine. BMC medical education. 2007;7(1):24.

6. Davis MH. Measuring individual differences in empathy: Evidence for a multidimensional approach. Journal of personality and social psychology. 1983;44(1):113.

7. De Waal FB, Preston SD. Mammalian empathy: behavioural manifestations and neural basis. Nature Reviews Neuroscience. 2017;18(8):498-509.

8. Strayer J, Roberts W. Children's anger, emotional expressiveness, and empathy: Relations with parents’ empathy, emotional expressiveness, and parenting practices. Social development. 2004;13(2):229-54.

9. Van Honk J, Schutter DJ, Bos PA, Kruijt A-W, Lentjes EG, Baron-Cohen S. Testosterone administration impairs cognitive empathy in women depending on second-to-fourth digit ratio. Proceedings of the National Academy of Sciences. 2011;108(8):3448-52.

10. Domes G, Heinrichs M, Michel A, Berger C, Herpertz SC. Oxytocin improves “mind-reading” in humans. Biological psychiatry. 2007;61(6):731-3.

11. Baez S, Flichtentrei D, Prats M, Mastandueno R, García AM, Cetkovich M, et al. Men, women… who cares? A population-based study on sex differences and gender roles in empathy and moral cognition. PloS one. 2017;12(6):e0179336.

12. Cunico L, Sartori R, Marognolli O, Meneghini AM. Developing empathy in nursing students: a cohort longitudinal study. Journal of clinical nursing. 2012;21(13-14):2016-25.

---

## [Editor Report · Decision Letter 1]

6 Sep 2021

Empathy and its associations with age and sociodemographic characteristics in a large UK population sample

PONE-D-21-14791R1

Dear Authors,

We’re pleased to inform you that your manuscript has been judged scientifically suitable for publication and will be formally accepted for publication once it meets all outstanding technical requirements.

Kind regards,

Marcel Pikhart

Academic Editor

PLOS ONE
---

## [Editor Report · Acceptance letter]

9 Sep 2021

PONE-D-21-14791R1 

Empathy and its associations with age and sociodemographic characteristics in a large UK population sample 

Dear Dr. Sommerlad:

I'm pleased to inform you that your manuscript has been deemed suitable for publication in PLOS ONE. Congratulations! Your manuscript is now with our production department. 

Kind regards, 

on behalf of

Dr. Marcel Pikhart 

Academic Editor

PLOS ONE